# Impact of the composition of feature extraction and class sampling in medicare fraud detection

Akrity Kumari
mit2020087@iiita.ac.in
Indian Institute of Information Technology Allahabad
Prayagraj, Uttar Pradesh, India

Narinder Singh Punn
pse2017002@iiita.ac.in
Indian Institute of Information Technology Allahabad
Prayagraj, Uttar Pradesh, India

Sanjay Kumar Sonbhadra
sanjaykumarsonbhadra@soa.ac.in
Shiksha 'O' Anusandhan
Odisha, Bhubaneswar, India

Sonali Agarwal
sonali@iiita.ac.in
Indian Institute of Information Technology Allahabad
Prayagraj, Uttar Pradesh, India

## ABSTRACT

With healthcare being critical aspect, health insurance has become an important scheme in minimizing medical expenses. Medicare is an example of such a healthcare insurance initiative in the United States. Following this, the healthcare industry has seen a significant increase in fraudulent activities owing to increased insurance, and fraud has become a significant contributor to rising medical care expenses, although its impact can be mitigated using fraud detection techniques. To detect fraud, machine learning techniques are used. The Centers for Medicaid and Medicare Services (CMS) of the United States federal government released "Medicare Part D" insurance claims are utilized in this study to develop fraud detection system. Employing machine learning algorithms on a class-imbalanced and high dimensional medicare dataset is a challenging task. To compact such challenges, the present work aims to perform feature extraction following data sampling, afterward applying various classification algorithms, to get better performance. Feature extraction is a dimensionality reduction approach that converts attributes into linear or non-linear combinations of the actual attributes, generating a smaller and more diversified set of attributes and thus reducing the dimensions. Data sampling is commonly used to address the class imbalance either by expanding the frequency of minority class or reducing the frequency of majority class to obtain approximately equal numbers of occurrences for both classes. The proposed approach is evaluated through standard performance metrics such as F-measure and AUC score. Thus, to detect fraud efficiently, this study applies autoencoder as a feature extraction technique, synthetic minority oversampling technique (SMOTE) as a data sampling technique, and various gradient boosted decision tree-based classifiers as a classification algorithm. The experimental results show the combination of autoencoders followed by SMOTE on the LightGBM (short for, Light Gradient Boosting Machine) classifier achieved best results.

## CCS CONCEPTS

• **Computing methodologies → Knowledge representation and reasoning**.

## KEYWORDS

GBDTs, Fraud, SMOTE, Autoencoders, Medicare, LightGBM

**ACM Reference Format:**
Akrity Kumari, Narinder Singh Punn, Sanjay Kumar Sonbhadra, and Sonali Agarwal. 2022. Impact of the composition of feature extraction and class sampling in medicare fraud detection. In *epiDAMIK 2022: 5th epiDAMIK ACM SIGKDD International Workshop on Epidemiology meets Data Mining and Knowledge Discovery, August 15, 2022, Washington, DC, USA.* ACM, New York, NY, USA, 10 pages.

## 1 INTRODUCTION

Fraud is described as the misuse of a company's system that does not always result in direct legal consequences. Frauds are dynamic and have no patterns. Out of various categories of fraud, insurance fraud is one of the subtypes, which is committed frequently. Insurance fraud is defined as any action performed to obtain a false insurance claim. There are again various categories of insurance fraud, one of which is healthcare insurance fraud that is committed in the healthcare industry. Fraud in healthcare insurance could be committed through the claimant (insured person) or the provider(doctor).

The fraudulent activities by providers involve:

- Charging for more expensive services than were actually given,
- Providing and thereafter charging for non-medically necessary treatments,
- Scheduling additional visits for patients,
- Recommending patients to other doctors when they do not seek further treatment,
- Phantom billing i.e. demanding a fee for services that were not rendered,
- Ganging, i.e. demanding a fee for services provided to members of family or other people accompanying the patient who did not receive any treatment for themselves.

Whereas, fraud by claimants involve:

- Claims on behalf of members and/or dependents who are not eligible,
- Modifications to membership forms,

- Hiding pre-existing conditions,
- Other coverage not disclosed,
- Prescription medication fraud,
- Failure to disclose claims arising from work-related injuries.

Thus, in the health insurance program, both beneficiaries, as well as healthcare providers, send false insurance claims to insurance companies in order to benefit from reimbursements. Such fraudulent activities by individuals or groups impact the lives of many innocent people as the beneficiaries have to pay higher insurance premiums rate for the services being received. Thus, insurance fraud creates a serious issue, hence, governments, insurance companies, and other organizations make an effort to discourage these kinds of activities by identifying the maximum number of fraudsters. To minimize such fraudulent activities, fraud detection is needed. The traditional technique of fraud detection, which is still commonly employed today, is detecting fraud patterns that have previously been encountered. This approach is primarily focused on the application of pre-established business rules (basic or advanced) to past data, which is insufficient given a large number of insurance claims and the wide range of fraudulent patterns. This system also necessitates continuous supervision by the expert in order to keep the rules up to date.

In recent years, several advancements are introduced in healthcare domain [27] with machine learning and deep learning technologies [4, 22, 25, 26, 30]. Machine learning methods have been applied to various other fraud detection problems such as tax fraud detection, credit card fraud detection, and bankruptcy fraud detection, and thus, are also being applied to fraud detection in healthcare insurance claims. Models that are build on machine learning and artificial neural networks (ANN) have made it possible to automatically extract features and build patterns and hence detect fraudulent activities more effectively and efficiently. Medicare [3] is a government-run health insurance program that covers approximately 54.3 million people in the United States. It covers persons over the age of 65 as well as younger people with specified medical conditions and disabilities. In Medicare program, insurance companies receive a huge number of requests for payment for the services provided by healthcare providers to their patients. Such requests are called insurance claims and a part of these might be fraudulent.

Healthcare is one of the vulnerable areas for perpetrators of fraud and owing to the continuous increase in fraud, waste, and abuse (FWA) activities in medicare programs necessitate the need for a fraud detection system to prevent the possible fraudulent activities arising in the Medicare program. This paper proposes an efficient framework using machine learning techniques to detect Medicare fraud. Following are the techniques that have been applied in building the framework's architecture:

- Feature extraction: This work use autoencoders as a feature extraction technique to minimise the number of features in the dataset by generating new ones from old ones.
- Class imbalance: This work uses SMOTE for handling the imbalanced ratios of output classes since non-fraudulent class constitutes a significant part of the dataset.
- Classification: This work trains various implementations or improvements of gradient boosted decision tree classifiers. A comparative study is done between various gradient

boosting algorithms like Catboost, XGBoost, AdaBoost, and LightGBM to get the best performing classifier on the medicare dataset.
- Analysis: The research work evaluates the performance of classifiers with F1-score and AUC metrics, and result shows that classifiers built with a combination of autoencoders and SMOTE attain better results.

The rest of the paper is divided into several sections. Section 2 describes background and related work behind the research work. Section 3 contains the architecture of the framework adopted to conduct the study. Section 4 describes the results and output of the algorithm. Finally, the concluding remarks are presented in Section 5.

## 2 BACKGROUND AND RELATED WORK

There exists a body of research for the application of machine learning in the domain of anomaly detection. Thus, this work started over various publications in the area of anomaly detection and led to the study of fraud detection in the healthcare domain [23], [17], [5], [32], [6], [31]. The previous publications on fraud detection in healthcare insurance claims and other related areas [28], [8] lead us to the question of whether various recent GBDTs implementations, along with autoencoder's automatic feature extraction capability to address high dimensionality, followed by class sampling, is a suitable algorithm for Medicare fraud detection.

### 2.1 Handling imbalanced dataset with the sampling method

The primary challenge of applying machine learning models in fraud detection, especially for medicare data is the highly imbalanced distribution of two classes: normal and fraudulent providers. When there is an imbalanced distribution of classes in a dataset, such as when the negative class (majority class) has a large number of data points in comparison to the positive class (minority class), class imbalance occurs (minority class). They usually give incorrect results and can be misleading with too optimistic scores if accuracy measures are taken into account. One of the reasons for these failures is that minority class points are seen as outliers that contain no information and be inclined toward majority class.

In order to address class imbalance, different training strategies can be used such as resampling (oversampling and undersampling), membership probability thresholding, and cost-sensitive learning. One of the most popular methods for dealing with an imbalanced dataset is to resample the data. Undersampling and oversampling are the two most common strategies that comes under resampling. Majority of the studies concerning the fraud detection in medicare datasets have used resampling techniques (usually, undersampling technique by varying the sampling ratios) to overcome the imbalanced class problem [7, 20, 21, 24, 34]. These studies have come to a conclusion that undersampling (down-sampling) is more efficient than oversampling as adding new data samples results in overfitting and increases in training time of the classifier, so this work attempts to discover whether an oversampling technique is a good proposal for addressing class imbalance problem. Therefore, this work proposes a method based on the oversampling technique known as SMOTE where the artificial samples are created for the minority

class. This technique helps to avoid the overfitting problem caused by random oversampling, which involves adding exact replicas of minority instances to the original dataset, whereas SMOTE employs a subset of data from the minority class as an example and then creates new synthetic identical instances.

*2.1.1 SMOTE.* This method synthetically increases the minority class by generating fresh examples of the minority class using specialized methods like the nearest neighbor and Euclidean distance. To generate fresh "synthesized" instances, the technique gives a set of simple rules. The created data is never an exact clone of one of its parents, despite the fact that each new synthetic data is built from its parents. There is no loss of essential information in SMOTE in contrast to undersampling.

To implement SMOTE, a library called imblearn is used that implements 85 variants of the SMOTE technique. Imbalanced-learn (also known as imblearn) is an open-source, MIT-licensed library that uses scikit-learn (also known as sklearn) and provides tools for dealing with imbalanced class categorization. It was first introduced by Chawla et al. [9].

## 2.2 Handling heterogeneous datasets with the dimensionality reduction method

To achieve valuable characteristics and accurate outcomes, machine learning models tend to incorporate as many features as feasible at the beginning. However, as the number of characteristics increases, the model's performance begins to deteriorate. Curse of dimensionality is a term used to describe this problem, which can lead to overfitting. Dimensionality reduction is the process of obtaining a set of principal features that reduces the dimensionality of the feature space in consideration with the aim that lower dimension representation retains some meaningful characteristics of original instances of the dataset. Dimensionality reduction is commonly applied when the dataset contains a large number of features and the medicare dataset contains 1360 attributes after one-hot encoding of the categorical variables necessitating the need for a dimensionality reduction step.

The two primary approaches to dimensionality reduction are feature extraction and feature selection. Feature selection is the process of finding the subset of features from the original features. Feature extraction is the process of creating new features from the existing feature of higher dimensional space to lower feature subspace. It is used to compress the data. This research work uses the feature extraction technique.

*2.2.1 Feature extraction.* Feature extraction is used to reduce the number of features in a dataset by creating a new set of features from the original set of features, afterwards the original set of features is removed. These new minimized sets of features should then be able to contain the maximum amount of information present in the actual features. In such a manner, a summarized version of the actual features can be created from a combination of the original set. Regardless of the difficulty with imbalanced datasets in Medicare, these also have a significant number of features that must be handled. A fraud detection system that is built using all features is usually not very efficient because the machine learning algorithms are impacted by insignificant or non-trivial features

during the training process leading to overfitting.
The reason for introducing the feature extraction are as follows:

- It produces better results than applying machine learning algorithms to original data, i.e boosts the classification scores.
- It reduces the memory and computation load on the hardware resources.
- It allows for easier visualization of data.
- It provides a deeper understanding of the fundamental structure of the data.
- It also reduces overfitting by the classifier.

To address the aforementioned issues, this research employs a nonlinear dimensionality reduction technique known as stacked autoencoder to generate robust and discriminative features for fraudulent instances, which will aid in the effective detection of fraudulent providers by grouping them into homogeneous clusters. Various alternative feature extraction strategies, such as early attempts to build on the projection method and involving mapping of input attributes in the original high-dimensional space to the new low-dimensional space with little information loss, have also been investigated. The two most well-known projection techniques are principal component analysis(PCA) and linear discriminant analysis (LDA). These techniques have been applied to anomaly detection in recent papers [10–13]. However, there are disadvantages associated with such projection techniques. The main drawback of the aforementioned approaches is that they perform linear projection among features while autoencoders can model complex, non-linear functions. Another drawback of these projection techniques is that most of these works tend to map data from high-dimensional to low-dimensional space by extracting features once, rather than stacking them to build deeper levels of representation gradually. Using artificial neural networks, autoencoders compress dimensionality by reducing reconstruction loss.As a result, it is simple to stack autoencoders by adding any number of hidden layers with the sequential API of the Python library. This gives the autoencoder the ability to extract meaningful features.

*2.2.2 Autoencoders.* Autoencoders are a special type of feedforward neural network in which input and output are the same. They compress the input into a lower-dimensional representation or code, that is used afterward to reconstruct the output. The code which is a condensed "summary" or "compression" of the input, is also known as the latent space representation. The representation obtained from the autoencoders has the following characteristics:

- They are data-specific which means they could only compress data that is identical to what the training was done on. This is in contrast to the MPEG-2 Audio Layer III (MP3) compression method, which only makes assumptions about "sound" in general, not specific sorts of sounds. Because the features it learns are face-specific, an autoencoder trained on photographs of faces would do a bad job compressing pictures of trees.
- They are lossy, which implies that when compared to the original inputs, the decompressed outputs will be degraded (similar to MP3 or JPEG compression). This is not to be confused with lossless arithmetic compression.

- They are automatically learned from data points, which is a useful property because the autoencoder makes it simple to train specialized instances of the algorithm to perform efficiently on a particular kind of input. It does not necessitate any new engineering, but it does necessitate appropriate training data.

An autoencoder is comprised of three parts: encoder, code, and decoder. The encoder compresses the input and generates the code, which the decoder subsequently uses to reconstruct the input. Building an autoencoder requires three components: an encoding function, a decoding function, and a distance function to calculate the amount of information loss between the compressed representation and the decompressed representation of the data (i.e. a "loss" function). The encoder and decoder are chosen to be parametric functions (generally, fully-connected feedforward neural networks, specifically the ANNs) that are differentiable with reference to the distance function, and allow the parameters of the encoding or decoding functions to be optimized using Stochastic Gradient Descent to minimize the reconstruction loss. Code is a single layer of an ANN with our desired dimensions. Before training the autoencoder, the number of nodes in the code layer (code size) and encoder-decoder layer is set as a hyperparameter. For the code generation, the input is first passed through the encoder, which is a fully-connected ANN. The output is subsequently generated solely using the code by the decoder, which has also a structure similar to ANN. The purpose is to get an output that is exactly the same as the input. The architecture of the decoder is usually identical to that of the encoder.

## 2.3 Choosing among various classification algorithms

Prokhorenkov et al. state that ensembles of gradient boosted decision trees (GBDT algorithms) are suitable for operating on heterogeneous datasets [19]. Heterogeneous data include features from a wide range of data types, i.e. from numerical to categorical features. Tabular datasets are frequently heterogeneous, and CMS's medicare claims data is an example of heterogeneous data. Khoshgoftaar et al. [19] show that CatBoost, LightGBM, and XGBoost, which are recent GBDTs implementations, are robust classifiers for highly imbalanced, insurance claims data. As a result of these findings, the current study examines the performance of four different types of GBDT algorithms (i.e. XGBoost, AdaBoost, CatBoost, and Light-GBM) on Medicare claims data. This study mainly explores which GBDT improvement performs the best on the Medicare dataset. For all of the GBDTs classifiers, hyper-parameters are near to default values, allowing for a fair baseline comparison.

*2.3.1 Gradient Boosted Decision Trees.* Gradient boosting is a technique for improving the performance of a machine learning model by using an ensemble (i.e. combination) of weak learners. On each problem, the actual performance of boosting methods is clearly influenced by the input and the weak classifier. Decision trees, specifically Classification and Regression (CART) trees, are usually the weak learners. A better prediction model is created by combining the output of several base learners. The class with the most

votes from weak learners could be the final result of the classification task. For gradient boosting methods, weak learners work in a sequential order. Each model aims to reduce the mistake introduced by the previous model. Trees in boosting-based classifiers are weak learners, but by stacking multiple trees in a row, each concentrating on the preceding model's errors, boosting algorithms become a very efficient and accurate model. To determine the errors, a loss function is utilised. For example, mean squared error (MSE) can be used for regression tasks, while logarithmic loss (log loss) can be used for classification tasks. When a new tree is introduced to the ensemble, the current trees do not change.

The steps involved in the boosting process are:

(1) Create a primary model with the input data,
(2) Make predictions on the entire dataset,
(3) Using the predictions and the actual values, calculate error,
(4) Give more weight to the wrong predictions,
(5) and create a new model that tries to rectify errors from the previous model,
(6) Make predictions on the whole dataset with the newly created model,
(7) Create a number of models with each model aiming at rectifying the errors generated from the previous model,
(8) Get the final model by weighting the mean of all the models.

The various boosting algorithms present in machine learning and used in this work are as follows:

- AdaBoost: AdaBoost (Adaptive Boosting) is a Machine Learning approach that is utilised as part of an Ensemble Method. It's quick, straightforward, and simple to programme. It does not have any tuning parameters. Decision trees with one level, or Decision trees with only one split, are the most popular algorithm used with AdaBoost. Decision Stumps is another name for these trees. This algorithm creates a model by giving all data points the same weight. It then gives points that are incorrectly categorised a higher weight. Then, in the following model, all of the points with greater weights are given more relevance. It will continue to train models until a smaller error is received [35].
- XGBoost: eXtreme Gradient Boosting, sometimes known as XGBoost, is a scalable machine learning approach based on tree boosting. It also employs a collection of weak decision trees. It's a linear model that uses parallel calculations to train trees. The following are the model's primary algorithmic implementation features:

(1) Sparse Aware implementation with automatic handling of missing data values.
(2) A block structure supports the parallelization of tree construction.
(3) Continued training to improve a model that has already been fitted using new data.

- LightGBM: Light Gradient Boosting Machine [16], is a decision tree-based gradient boosting architecture that improves model efficiency while reducing memory utilisation. It employs two innovative techniques: Gradient-based One Side Sampling (GOSS) and Exclusive Feature Bundling (EFB), which address the shortcomings of the histogram-based algorithm utilised in all GBDT frameworks. GOSS and EFB are

two strategies that make up the LightGBM algorithm, and they work together to make model run smoothly and provide a competitive advantage over competing GBDT frameworks.

- CatBoost: Categorical Boosting [15], an open-source gradient boosting machine learning algorithm. Ordered Target Statistics and Ordered Boosting are two of the advances used. CatBoost is well-suited to machine learning tasks involving category, heterogeneous data as a Decision Tree-based method [18]. It produces good results without extensive data training and with a small amount of data.

Bauder et al. [5] conducted extensive studies on fraud detection utilizing both supervised and unsupervised learning techniques for fraud detection. Their experimentation was based on Medicare Part B provider data and applied methods for detecting outliers. They then merged a number of Medicare-related datasets. The combined medicare dataset was labeled with the LEIE data. Their work also takes into account the data imbalance problem, by using various data levels as well as algorithm level approaches. According to their research, the data level performed better In comparison to the algorithmic level.

Johnson and Khoshgoftaar [16] studied the performance of various deep learning algorithms on the Medicare fraud detection challenge [18, 29]. The authors broaden the scope of sample procedures while altering the learners' classification thresholds. They showed that a hybrid strategy of random undersampling and random oversampling has an effect on the AUC of deep learning algorithms. It was found that GBDT algorithms show promising performance in the task of Medicare Fraud detection. For most experiments, they found CatBoost is the strongest performer when Random Undersampling was used for class balancing, the ratio was 1:1 for the minority to majority class. According to their research, the best performance was obtained for the non-aggregated Medicare Part B and Part D datasets.

From all the research work covered so far, it is observed that researchers have yet not compared the performance of CatBoost, LightGBM, XGBoost, and AdaBoost along with autoencoder's feature extraction capability on the work of identification of fraud in Medicare data. The Medicare data used in this study have various categorical features, for example, Drug Name, Provider State, and Specialty Description. Out of these, the Drug name feature has 1193 distinct values in the dataset. To represent this high cardinality feature in the building model, one-hot encoding is applied which increases the dimensionality of the dataset considerably. To address this curse of dimensionality, autoencoders are used to reduce dimensionality. By doing so, the inclusion of high cardinality categorical features like the drug name became easier during the training of models. Hence, this research work takes the advantage of the autoencoder's automatic feature extraction capability to contribute to the body of research area that these studies relate to. This research work compares four different types of GBDT algorithms, for the task of medicare fraud detection. We also address the class imbalance problem by the means of SMOTE technique, which is still unexplored in the medicare dataset.

## 3 IMPLEMENTATION OF THE FRAMEWORK

The implementation of the proposed framework covers the following stages:

1. Data collection and preparation,
2. Feature selection and feature engineering,
3. Choosing the machine learning algorithm and training our model,
4. Evaluating our model

The structure of the framework is shown in Fig. 1. The techniques applied in building the framework consist of autoencoder as feature extractor, SMOTE for data sampling, and GBDTs as a classifier.

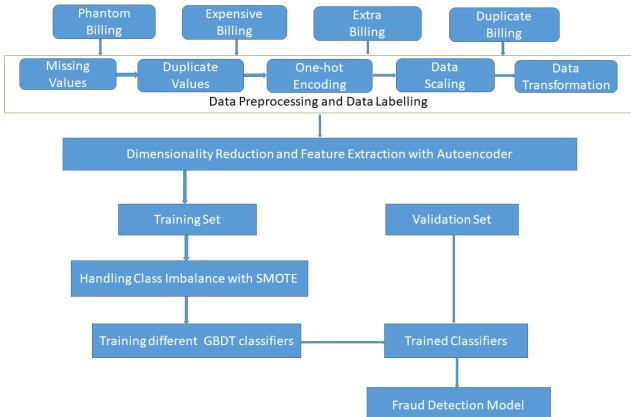

**Figure 1: Framework for the fraud detection.**

## 3.1 Data collection and preparation

This paper uses the Medicare Part D insurance claims dataset [1]. CMS provides a number of publicly accessible files every year, and this dataset is one of them. These files are combinedly known as Medicare Provider Utilization and Payment Data: Physician and Other Supplier and Medicare Provider Utilization and Payment Data: Part D Prescriber (Part D). This data is present in character separated value (CSV) format, and there is one file for each year, this work uses the file for the year 2018. CMS prepares a document that specifies all the features or attributes of the Medicare Part D insurance claims data. Each row in the dataset describes a provider, mainly by the national provider identifier (NPI) of the provider, and secondarily by various features that provide information related to the name, demographics, and location (state and city) of the provider. This dataset contains information about the drug names they prescribe to their patients. It also contains records related to provider type indicating the nature of the provider's practice, such as ophthalmology, family practice, nursing, and so forth. For every drug prescription, the provider has submitted a drug cost claim to Medicare, and there is one record in the PUF file for the year. For each row, along with the drug name, there are certain aggregate statistics associated with it, namely the total number of unique Medicare Part D beneficiaries with at least one claim for the drug, and the aggregate drug cost paid for all associated claims. The features of the dataset used in the experiment are discussed in Table 1.

**Table 1: Features used for the experiment.**

| Name | Description | Type |
|---|---|---|
| npi | Unique identification of provider, used for labeling | Numeric |
| nppes_provider_state | The state where the provider is located | Categorical |
| speciality_description | Medical provider's specialty (or practice) | Categorical |
| description_flag | A flag that indicates the source of the specialty_description | Categorical |
| drug_name | The name of the drug prescribed (or filled). | Categorical |
| bene_count | The total number of unique Medicare Part D beneficiaries with at least one claim for the drug | Numeric |
| total_claim_count | Number of Medicare Part D Claims, including refills | Numeric |
| total_30_day_fill_count | Number of standardized 30-day fills, including refills | Numeric |
| total_day_supply | Number of day's supply for all claims | Numeric |
| total_drug_cost | Aggregate cost paid for all claims. | Numeric |

The second dataset used in this study is the List of Excluded Individuals and Entities (LEIE) data [2]. The LEIE dataset is updated by the Office of Inspector General (OIG) on a monthly basis. This file is also present in a CSV format. It contains information about the healthcare providers that are prohibited from sending claims to Medicare because they have previously broken Medicare's rules and regulations for submitting claims. In this study, the important attributes of the LEIE dataset are the NPI and the exclusion type. Initially, the part D dataset is unlabelled. This work derives a label for the Medicare Part D data from the LEIE data on the basis of the NPI attribute. If an NPI from the Medicare Part D data is present in the LEIE data, then all Medicare Part D data, having that NPI is labeled as fraudulent. To provide labels, the Part D dataset and LEIE dataset are merged using left join on NPI, and all the NAN records obtained after combining the datasets are labeled as non-fraudulent (class 0).

## 3.2 Data preprocessing

All the required data pre-processing steps such as handling missing and duplicate values, data scaling, data transformation, and data filtering are performed on the dataset. Following are the two significant issues confronted in the preprocessing step of this study.

*3.2.1 Handling Heterogenous datasets.* Since the medicare dataset is a heterogeneous dataset(i.e. consisting of both categorical and numerical features), the features like specialty_description, nppes_provider_state and drug_name are categorical in nature, which cannot be used with the classifiers in their raw form. Table 2 shows the counts of distinct values of categorical features in the dataset. So, the data need to be processed in order to convert the categorical features into numerical features to train the model with the GBDTs classifier. Hence, the categorical features were encoded by one-hot encoding. But performing this increased the dimensionality of data drastically. Autoencoder was then applied to reduce the dimension. The implementation uses Keras's deep learning framework, which is a functional API of Python on the top of TensorFlow to build the autoencoder. An autoencoder was designed with the following

**Table 2: The cardinality of categorical features in dataset.**

| Features | Distinct Values |
|---|---|
| specialty_description | 101 |
| nppes_provider_state | 59 |
| description_flag | 2 |
| drug_name | 1160 |

architecture, two hidden layers in the encoder as well as the decoder. The hyperparameters set before training an autoencoder to get good results are as follows:

- Code size: It is defined as the number of nodes in the middle layer or code layer. More compression occurs when the size is smaller. Its value is set to 32 during experimentation.
- The number of layers: The depth of the autoencoder depends on the performance requirement. The architecture of the autoencoder in the experiment consists of 2 layers in the encoder as well as the decoder, without taking into consideration input and output.
- The number of nodes per layer: Since the layers are placed one after another, the autoencoder architecture in the implementation is a stacked autoencoder. Stacking autoencoders usually resemble a "sandwich." With each consecutive encoder layer, the number of nodes per layer drops and then increases in the decoder. The decoder is symmetric to the encoder in terms of the layer structure. The number of nodes is set to 100 in the first layer and to 50 nodes in the second layer. The activation function used is relu in the encoder layer and tanh in the decoder layer.
- Loss function: mean squared error (mse) or binary cross-entropy can be used by the autoencoder. When the input values are in the range [0, 1], cross-entropy is used, otherwise, mean squared error is used. This work uses mse as loss function and adam as optimizer.

To fit the model, the epoch value is set to 50 with early stopping to avoid overfitting, and the batch size is set to 128. The model is fitted to all non-fraud instances in the dataset to prevent the autoencoder from merely learning to replicate the inputs to the output, that is, without any meaningful representations being learned. In the implementation, three layers of autoencoders are stacked for building the final model.

*3.2.2 Handling Imbalanced dataset.* The medicare dataset is highly imbalanced in nature. The non-fraudulent class overwhelms the majority of the data. Fraudulent transactions comprise 341 instances or 0.055%, thus the dataset is highly imbalanced with respect to the majority to minority classes. It is necessary to deal with the class imbalance problem present in the medicare dataset. To deal with this problem, this research work applies a data sampling method called SMOTE. The minority class i.e fraudulent class is oversampled by adding synthetic minority samples and making the minority to majority class proportion the same. By performing the class balancing step, class 1 would be learned as much as class 0. The SMOTE approach is used only on the training dataset to ensure that the classification algorithm fit the data adequately.

*3.2.3   Choosing the machine learning algorithm and training our model.* The latent space representation is obtained from the autoencoder's feature extraction technique ability. These extracted features are then used to train different classification models. In this work, various GBDTs implementations are trained on the extracted feature or latent space representation to obtain the best performing classifier with regard to F1-Score and AUC score with the aim to accurately predict fraud or non-fraud outcome for the data points of which class label is not known. For this research work following classification algorithms are used: XGBoost, CatBoost, AdaBoost and LightGBM. To provide a fair baseline comparison the hyperparameters for all the classifiers are set to default values.

*3.2.4   Evaluating the model.* This study evaluates the impact of employing the data sampling technique (SMOTE) and feature extraction technique (autoencoders) on classifier performance. The implementation is divided into four sections, first is the use of SMOTE only, second is the use of feature extraction only, third is the use of both SMOTE and feature extraction, and the last one is a baseline (no SMOTE and no feature extraction). When used with the SMOTE preprocessing phase, the feature extraction preprocessing step is executed first in the experiment.

For all the learners stated above, all of them produced an accuracy score greater than 85%. But the problem with fraud detection is that it has a skewed distribution for the target class. Therefore the accuracy metric is always misleading. The purpose of such research is to see how successfully each fraud and non-fraud class is classified. The percentage of precision and recall (count of true positives, false positives, true negatives, and false negatives) are the important metrics to be considered. Precision shows the percentage of non-fraudulent classes labeled as a fraud, while recall shows the percentage of fraudulent classes classified as non-fraudulent, which is even more dangerous in the task of fraud detection. So, the primary metric for evaluation in this study is the F1-score, which is the harmonic mean of precision and recall and takes into account both metrics and is a more considerable indicator for datasets with a high-class imbalance ratio. Table 1 shows a comparison between the above-mentioned classification algorithms in terms of various performance metrics. It depicts the accuracy, precision, recall, F1-score, and AUC score for each learner. Comparing all the aforesaid algorithms, LightGBM produces better results concerning AUC and f1-score rate.

# 4   EXPERIMENTAL RESULTS

## 4.1   Metrics for Evaluation of classifier Performance

To evaluate the proposed systems, standard performance metrics are used to calculate the performance of the system (accuracy, precision, recall, f-score, and AUC score).

*4.1.1   Accuracy.* In a classification problem, the accuracy score is defined as the ratio of the number of correct predictions to the total number of instances.

$$\text{Accuracy Score} = \frac{\text{Number of correct predictions}}{\text{Total number of instances}} \quad (1)$$

But, this prediction score is unreliable for an unbalanced distribution of classes or skewed dataset because the training and

**Figure 2: Confusion Matrix.**

evaluation as per this measure create a model that is likely to predict the non-fraud class (majority class) for all the test examples by increasing the percentage of True Negative and thus, the value rises to 99%. Hence, the confusion matrix is preferred for evaluating the model, which is a summary of correct and incorrect prediction values compared with the actual values of the input data, divided among classes as shown in Fig. 2, where TP (true positive) and TN (true negative) are correct predictions and FP (false positive) and FN (false negative) are wrong predictions. In terms of TP, TN, FP, and FN, accuracy is calculated as shown in Eq. 2

$$Accuracy = \frac{TP + TN}{TP + TN + FP + FN} \quad (2)$$

*4.1.2   AUC-Score.* The Receiver Operator Characteristic (ROC) curve is a statistic for evaluating binary classification issues. It's a probability curve that plots the TPR against the FPR at various threshold levels, allowing the signal to be distinguished from the noise. The AUC is a summary of the ROC curve that assesses the ability of a classifier to distinguish between classes. The AUC measures how successfully a model can distinguish between positive and negative classifications. The higher the AUC number, the better.

*4.1.3   Precision.* Precision indicates how many of the instances that were predicted positively by the model turned out to be actually positive. It is calculated as shown in Eq. 3.

$$Precision = \frac{TP}{TP + FP} \quad (3)$$

*4.1.4   Recall.* Recall indicates how many of the actual positive cases the model is able to correctly predict. It is computed using Eq. 4.

$$Recall = \frac{TP}{TP + FN} \quad (4)$$

*4.1.5   F1-Score.* The harmonic mean or weighted average of Precision and Recall is the F1-score (Eq. 5). Both false positives and false negatives are taken into account in this score.

$$F1 = \frac{2 * precision * recall}{precision + recall} \quad (5)$$

## 4.2   Results

The technique of t-distributed stochastic neighbour embedding (TSNE) [33] is used to visualise transaction data. TSNE is a statistical method for visualising high-dimensional data by assigning a two- or three-dimensional map to each datapoint. The method is a simplified form of Stochastic Neighbor Embedding, and it improves graphics by reducing the tendency for points to cluster in the map's

**Table 3: Scores for various classifier with Feature Extraction and SMOTE.**

| Classifier | Feature extraction | Data Sampling | Precision | Recall | F1-score | AUC | Accuracy |
|---|---|---|---|---|---|---|---|
| Catboost | none | none | 0.0000 | 0.0000 | 0.0000 | 0.4999 | 0.9995 |
| | Autoencoders | none | 0.6582 | 0.6117 | 0.6341 | 0.7879 | 0.9282 |
| | none | SMOTE | 0.2096 | 0.5652 | 0.3058 | 0.7821 | 0.9988 |
| | Autoencoders | SMOTE | 0.8585 | 1.0000 | 0.9239 | 0.9860 | 0.9761 |
| AdaBoost | none | none | 0.0000 | 0.0000 | 0.0000 | 0.5000 | 0.9995 |
| | Autoencoders | none | 0.6129 | 0.4470 | 0.5170 | 0.7075 | 0.9150 |
| | none | SMOTE | 0.2631 | 0.5434 | 0.3546 | 0.8214 | 0.9981 |
| | Autoencoders | SMOTE | 0.6043 | 0.9882 | 0.7500 | 0.9392 | 0.9044 |
| XGBoost | none | none | 0.0000 | 0.0000 | 0.0000 | 0.5000 | 0.9995 |
| | Autoencoders | none | 0.7142 | 0.3529 | 0.4724 | 0.6684 | 0.9198 |
| | none | SMOTE | 0.0039 | 0.8378 | 0.0078 | 0.8425 | 0.8472 |
| | Autoencoders | SMOTE | 0.6439 | 1.0000 | 0.7834 | 0.9530 | 0.9197 |
| LightGBM | none | none | 0.0000 | 0.0000 | 0.0000 | 0.4994 | 0.9984 |
| | Autoencoders | none | 0.5955 | 0.6235 | 0.6091 | 0.7877 | 0.9186 |
| | none | SMOTE | 0.0692 | 0.8108 | 0.1276 | 0.9014 | 0.9920 |
| | Autoencoders | SMOTE | 0.9444 | 1.0000 | 0.9714 | 0.9950 | 0.9914 |

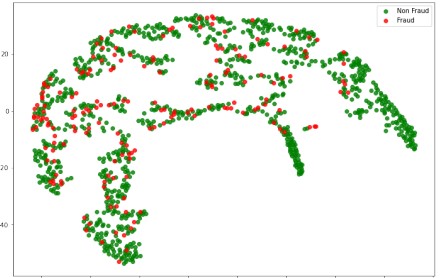

**Figure 3: Latent space representations before feature extraction using t-SNE projection.**

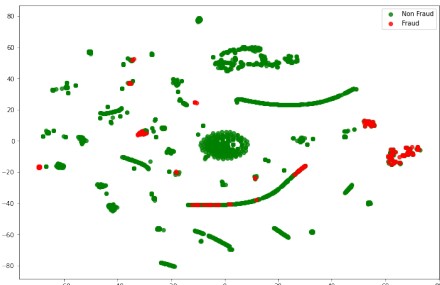

**Figure 4: Latent space representations after feature extraction using t-SNE projection.**

centre. t-SNE surpasses existing approaches when it comes to creating a single map that exhibits structure at several sizes. This is especially important for high-dimensional data that is scattered across multiple low-dimensional manifolds but is related to one another, such as images of items from various classes taken from diverse angles. Fig. 3 and Fig. 4 represent the scatter plot of the two dimensions before and after feature extraction from autoencoders, respectively.

Table 3 shows the values for precision, recall, F1-score, accuracy, and AUC values for each of the selected classifiers. Each row presents the outcomes of a unique and sequential combination of the first (feature extraction) and second (class balancing) preprocessing stages. The highest score for all the performance metrics and for each classifier is highlighted.

The combination of Autoencoder followed by SMOTE emerges to be the most effective. It has the highest F1-score (0.9714), the highest AUC (0.9950), the highest recall (1.0000), and the highest precision (0.9444) for the LightGBM classifier. Moreover, from the result, it is observed, that the baseline (no feature extractor and no class balancing) is performing the worst for all the mentioned classifiers. It yields the lowest recall, precision, F1-score, and AUC.

The F1-score incorporates both precision and recall, as earlier mentioned. As a result, the F1-score is given greater significance than its individual parts, precision, and recall. Therefore, based on the F1-score and AUC score values from Table 3, combinations of Autoencoders followed by SMOTE for LightGBM produces best result.

The experiments involving only autoencoders are also giving some better results because the latent representation is robust toward the imbalanced class due to the fact that the latent features extracted from the autoencoder have strong clustering power. The latent features allow the model to group the healthcare providers into clusters and make it easier to identify fraudulent behaviors, and this can be seen in Fig. 4 (fraudulent latent representations (i.e. red points) concentrate on separate clusters of latent space) [14].

### 4.3 Comparison with previous literature

According to the publications that address the same domain, this study outperformed the other results mentioned in Table 4. It produces better result than Shamitha et al. [29] because their study use PCA for dimensionality reduction which fails to capture the non-linear correlations between features but the current work use

**Table 4: Comparative Performance Analysis with previous work.**

| Research article | Precision | Recall | F1 | AUC | Accuracy |
|---|---|---|---|---|---|
| Shamitha et al. [29] | 0.9700 | 0.7300 | - | - | - |
| Hancock and Khoshgoftaar [16] | - | - | - | 0.7250 | - |
| This study | 0.9444 | 1.0000 | 0.9714 | 0.9950 | 0.9914 |

Autoencoder which can extract the non-linearly correlations between mutiple features and also perform dimensionality reduction. Besides, this study produces better result than Hancock and Khoshgoftaar [16] because their work relies only on catBoost's internal mechanism for encoding categorical features while this study empolys autoencoder for dimensionality reduction in addition to catboost classifier.

## 5 CONCLUSION

Healthcare being an integral component of people's lives has increased the requirement of health insurance schemes over the past few years. But, increasing insurance programs have motivated fraudsters to accomplish fraudulent activities on such schemes for their monetary gain. In an attempt to increase transparency and lessen fraud, there is a requirement for an efficient fraud detection system for the health insurance claims. To address this, an efficient framework is designed which deals with efficient solutions to eliminate problems associated with highly imbalanced and heterogeneous data. With exhaustive experiments using combination of several techniques such as data preprocessing, dimensionality reduction, oversampling and classifiers. Several learners are trained and compared to find the most effective one in building the fraud detection model. Among the classifiers under consideration, Light-GBM produced the best F1-score and AUC score when implemented with autoencoder followed by SMOTE technique. That is, applying feature extraction followed by data sampling outperformed the baseline architecture and produced better classification results. For further optimization, this work also performs L1-regularization and stacked various layers of the autoencoders, and the final goal of finding the best answer to the problem was fairly accomplished.

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
