# OpenReview forum: "Impact of the composition of feature extraction and class sampling in medicare fraud detection"
_ACM.org/SIGKDD/2022/Workshop/epiDAMIK — KDD 2022 Workshop epiDAMIK Poster_

### Official Review · Reviewer_Ns9d · 2022-06-21
**Impact of the composition of feature extraction and class sampling in Medicare fraud detection- Review**

**Rating:** 3
**Confidence:** 4

**Review:**

This paper elaborates on the feature extraction, data sampling, and application of various ML algorithms on the Medicare Part D Insurance Claims dataset to get improved performance. The proposed model first ingests 2 datasets, labels unlabelled columns, and imputes missing values with zeros. Then the categorical features are passed through One-Hot Encoding and then dimensionality reduction is performed using an Autoencoder. After that, SMOTE is used to oversample the minority class and the resultant data is fed into the CatBoost, AdaBoost, XGBoost, and LightGBM classifiers, and their resultant performance metrics are compared. The results of the best classifier are then compared to those of a couple of previous works.


Strong Points:
1. The work properly describes the dataset and relevant history to set the context of this work.
2. The work explicitly elaborates on the need to introduce feature extraction methods in a precise manner.
3. The work provides visualizations to represent the latent space representation before and after feature extraction.

Weak points:
1. In table 4, where the results of previous works are compared, some metrics of previous works have not been shown. Could they not have been replicated? Otherwise, it does not seem to be a fair comparison.
2. A major chunk of the paper is pretty well known among the ML community. For example, Section 4.1 could have easily been sent to the appendix, the first part of section 2.2 is pretty well known, and the last paragraph of section 2.1.1 could have just been sufficed with a simple citation.
3. Citations are missing in some places. eg, in the paragraph right after section 2, one citation is a "?".
4. Some acronyms have not been elaborated on. eg, GBDT, CART, etc.
5. A confusion matrix is defined, but results are not present anywhere.

Minor points:
1. Some places have grammatical errors, especially subject-verb-agreement.

---

### Official Review · Reviewer_bCPr · 2022-06-23
**The author of the paper titled "Impact of the composition of feature extraction and class sampling in Medicare fraud detection" demonstrated the impact of data sampling and feature extraction on the classification of fraud in medical data. The paper's Figures 3 and 4 are particularly intriguing and influential. To achieve publication level, I feel this work should contain a more robust analysis and explanation of the classification performance measure.**

**Rating:** 1
**Confidence:** 3

**Review:**


Pros:
1) The problem statement is compelling.
2) Figure 3 and Figure 4 are striking.
3) Results seem promising.

Cons:
1) Lack of novelty.
2) Why had just Autoencoder been utilised for feature extraction?
3) Why have the authors used solely SMOTE for data sampling?
4) Insufficient explicability in model architectures and outcomes.

---

### Official Review · Reviewer_Ss4d · 2022-06-25
**Machine learning based medical fraud detection that tackles class imbalance**

**Rating:** 3
**Confidence:** 4

**Review:**

Overview:
The paper deals with the binary classification task of medical fraud detection using financial, medical, and demographic features. They use autoencoders for automatic feature extraction and experiment with different Gradient Boosted Decision Tree algorithms. They also deal with severe class imbalance with the popular SMOTE method to generate more data from the minority class.

Strengths:
The paper is well-motivated and organized with an empirical justification of various parts of the modeling pipeline. The qualitative results of low-dimensional embeddings are also insightful in showing the usefulness of autoencoder approach.

Weakness:
The paper lacks technical novelty since they use well-known statistical methods for modeling and a simple neural autoencoder for feature extraction. Comparing with more recent ML and neural methods on tabular datasets [1,2] would have been useful.

[1] Gorishniy, Yury, et al. "Revisiting deep learning models for tabular data." Advances in Neural Information Processing Systems 34 (2021): 18932-18943.
[2] Borisov, Vadim, et al. "Deep neural networks and tabular data: A survey." arXiv preprint arXiv:2110.01889 (2021).